# Standardization of Somatic Variant Classifications in Solid and Haematological Tumours by a Two-Level Approach of Biological and Clinical Classes: An Initiative of the Belgian ComPerMed Expert Panel

**DOI:** 10.3390/cancers11122030

**Published:** 2019-12-16

**Authors:** Guy Froyen, Marie Le Mercier, Els Lierman, Karl Vandepoele, Friedel Nollet, Elke Boone, Joni Van der Meulen, Koen Jacobs, Suzan Lambin, Sara Vander Borght, Els Van Valckenborgh, Aline Antoniou, Aline Hébrant

**Affiliations:** 1Department of Clinical Biology, Jessa Hospital, 3500 Hasselt, Belgium; 2Department of Clinical Biology, University Hospital Antwerp, 2650 Edegem, Belgium; Marie.LeMercier@uza.be; 3Center for Human Genetics, University Hospital Leuven, 3000 Leuven, Belgium; e_lierman@yahoo.com; 4Laboratory of Molecular Hematology, Ghent University Hospital, 9000 Ghent, Belgium; Karl.Vandepoele@uzgent.be; 5AZ Sint-Jan Brugge-Oostende AV, Department of Laboratory Medicine, 8000 Brugge, Belgium; Friedel.Nollet@azsintjan.be; 6Department of Laboratory Medicine, AZ Delta Hospital, 8800 Roeselare, Belgium; elke.boone@azdelta.be; 7Molecular Diagnostics, Ghent University Hospital, 9000 Ghent, Belgium; Joni.VanderMeulen@UGent.be; 8Clinical Laboratory, AZ St-Lucas Hospital, 9000 Ghent, Belgium; Koen.Jacobs@AZSTLUCAS.BE; 9Department of Pathology, University Hospital Antwerp, 2650 Edegem, Belgium; Suzan.Lambin@uza.be; 10Laboratory of Pathology, University Hospital Leuven, 3000 Leuven, Belgium; sara.vanderborght@uzleuven.be; 11Cancer Centre, Sciensano, 1050 Brussels, Belgium; Els.VanValckenborgh@sciensano.be (E.V.V.); Aline.Hebrant@sciensano.be (A.H.); 12Department of Quality Laboratories, Sciensano, 1050 Brussels, Belgium; Aline.Antoniou@sciensano.be

**Keywords:** cancer, classification, guideline, NGS, variant

## Abstract

In most diagnostic laboratories, targeted next-generation sequencing (NGS) is currently the default assay for the detection of somatic variants in solid as well as haematological tumours. Independent of the method, the final outcome is a list of variants that differ from the human genome reference sequence of which some may relate to the establishment of the tumour in the patient. A critical point towards a uniform patient management is the assignment of the biological contribution of each variant to the malignancy and its subsequent clinical impact in a specific malignancy. These so-called biological and clinical classifications of somatic variants are currently not standardized and are vastly dependent on the subjective analysis of each laboratory. This subjectivity can thus result in a different classification and subsequent clinical interpretation of the same variant. Therefore, the ComPerMed panel of Belgian experts in cancer diagnostics set up a working group with the goal to harmonize the biological classification and clinical interpretation of somatic variants detected by NGS. This effort resulted in the establishment of a uniform, two-level classification workflow system that should enable high consistency in diagnosis, prognosis, treatment and follow-up of cancer patients. Variants are first classified into a tumour-independent biological five class system and subsequently in a four tier ACMG clinical classification. Here, we describe the ComPerMed workflow in detail including examples for each step of the pipeline. Moreover, this workflow can be implemented in variant classification software tools enabling automatic reporting of NGS data, independent of panel, method or analysis software.

## 1. Introduction

Over the recent years, most molecular diagnostic laboratories have replaced their single gene assays for analysing tumour samples for clinically relevant variants to massive parallel sequencing, commonly called Next-Generation Sequencing (NGS). This technique allows the simultaneous interrogation of multiple genes in several samples starting from limited amounts of DNA. This major change in diagnostics has now become time- and cost-efficient due to the improved target enrichment methods and the reduced cost per sample through the massive sequencing capacity [1].

Cancer development is a very complex process that requires the accumulation of several different acquired genetic changes, also called variants. These include drivers that are essential for tumour development, and passengers, which accidently occur [2]. Driver variants in oncogenes are activating changes mostly generated by missense variants or in-frame insertion/deletions (indels) at very specific positions called hotspots, or by gene amplifications or fusions resulting in a gain-of-function (GoF) of at least a function of that gene product, which often is its kinase activity. Driver variants in tumour suppressor (Ts) genes on the other hand negatively affect the proper function of that protein, resulting in a diminished or abolished protein activity or level. These variants thus induce a loss-of-function (LoF), and include nonsense variants, frameshift indels and splice site changes, as well as genomic aberrations that produce a null allele, such as a (partial) gene deletion. Occasionally, a change-of-function can also result in a cancer driver. Epigenetic inactivation can occur as well [3] but is not described here.

Although the number of clinically relevant tumour-associated variants is steadily increasing, it is still too low to justify whole genome or exome sequencing [4]. A targeted screening panel from a dozen to a few hundred genes is currently sufficient for clinical patient management and allows the detection of single nucleotide variants (SNV) and indels, but can also find copy number variations (CNV), tumour mutation burden (TMB), homologous recombination deficiency (HRD) and microsatellite instability (MSI) [5,6]. Moreover, exon skipping, gene fusions and changes in expression levels can also be detected when starting preferentially from the RNA of a tumour specimen [7].

Independent of the method, targeted NGS analysis ultimately results in a variant call format (Vcf) file for each sample containing a list of genomic positions at which a change was observed, compared to the human reference sequence. This list is then extensively filtered to yield a selected number of clinically relevant variants. These filtering steps typically depend on the underlying clinical question and will differ for the detection of somatic variants in a cancer biopsy versus germline variants in a recessive disease entity. After these filtering steps, the remaining variants then need to be classified for their tumour-inducing and diagnostic potential before reporting can take place. This crucial classification step includes (1) the biological classification, i.e., its predicted potential to induce or stimulate tumorigenesis through an altered protein function, and (2) the tumour type-dependent clinical interpretation, which relates to the potential to alter the clinical management of the patient. It is important to emphasize that the biological classification of a variant is independent of the underlying malignancy, whereas the clinical classification is strongly influenced by the specific pathology in which the variant is found, as well as by several other clinical data. For germline variants, guidelines have been described to classify variants in biological classes [8] as well as tier-based clinical interpretation classes [9,10,11]. For somatic variants however, only guidelines for the clinical interpretation have been extensively described [12,13,14] while those for the biological classification are only touched upon briefly [15] or described for specific disease conditions [16]. Both classifications are however necessary to be able to standardize variant calling in a diagnostic setting. Until now, lab-specific and thus different biological classification systems are being used, which can have a significant impact on the reported clinical consequence.

In a survey organized by the Belgian MolecularDiagnostic.be working group on the biological and clinical classification of somatic variants, critical differences in the biological classification were noticed with the expected impact on the clinical reports of the same patients. Very similar results were reported by the ACMG working group [12]. Therefore, we set up an expert group to harmonize this first process and present here the Belgian guidelines that were proposed for the biological classification of somatic variants detected by NGS. For clinical interpretation, the four tier-based classes reported by the ACMG were retained, which are described in detail elsewhere [12].

## 2. Results

### 2.1. Expert Panel

All molecular diagnostic laboratories in Belgium were invited to join an expert group with a focus on the analysis and interpretation of NGS variants in the field of solid and haematological tumours. A total of 91 experts, mainly consisting of molecular biologists, clinical biologists, oncologists and haematologists were selected to form the Commission for Personalised Medicine (ComPerMed) expert panel. Members represent 27 Belgian hospitals of which seven academic hospitals, 19 non-academic hospitals and one laboratory not associated with a hospital. In addition, members of the national institute for science and health (Sciensano, Brussels, Belgium) were represented to guide the project. The major aim of this working group is to install guidelines and workflows for personalized cancer treatment and optimized patient management (https://www.compermed.be/). One of the goals was to harmonize the biological classification of somatic variants in Belgium through well-defined workflows to avoid differences between laboratories. For that, a working group of molecular biology experts in cancer genomics was installed. The second goal was to streamline the clinical interpretation and reporting of these biologically classified somatic variants to ameliorate the inter-laboratory concordance of clinical data. The expert panel gathered the first time in January 2018, with subsequent meetings every 6 months and multiple intermediate electronic communications. The final document was obtained in January 2019 and published on the BELAC website in French (https://economie.fgov.be/sites/default/files/Files/Publications/files/Belac-FR/2–405NGS-FR.pdf) and Dutch (https://economie.fgov.be/sites/default/files/Files/Publications/files/Belac-NL/2–405NGS-NL.pdf) in July 2019. Note that these guidelines change to English from page five onwards. The focus was on those genes that were decided by the ComPerMed panel as required to be screened by NGS in the different tumour types [17,18,19,20]. These genes are listed in Table 1 and Table 2.

### 2.2. Variant Calling and Annotation

Targeted NGS is currently a well-established screening method in many molecular diagnostics laboratories that perform tumour screening for personalized clinical management. These panels most commonly include twenty to fifty genes, which make them cost-efficient and bioinformatically manageable. Moreover, small to medium-sized panels avoid the mass introduction of technical artefacts due to e.g., sequencing-, alignment- and analysis-related errors. A large number of NGS data analysis pipelines exist, but all will result in a Vcf file for each sample. In this Vcf file, the chromosome and genomic position for each variant is provided, as well as the reference and alternative base, the coverage at that position and its variant allele frequency (VAF). Each variant can be a substitution, deletion, insertion or duplication of one or more nucleotides compared to the human genome reference sequence. The Vcf file also includes quality-related parameters to be able to judge the likelihood of a correct call. Subsequently, these files are used for annotation, i.e., assessing the consequence of that change at the transcript and protein level, and its associated functional and subsequent clinical effect. For each variant, the HGVS nomenclature [21] at the transcript and protein level, including the reference sequence accession number with versioning is provided, e.g., *EGFR* NM_005228.3: c.2573T>G; NP_005219.2: p.(Leu858Arg), or *TET2* NM_001127208.2: c.419del; NP_001120680.1: p.(Asn140Metfs*5). Moreover, annotation programs include additional information such as its presence in population databases (gnomAD, dbSNP, 1000 genomes) and/or disease-associated databases (COSMIC, CIVIC, ClinVar, etc.) as well as the outcome of functional prediction programs (SIFT, MutationTaster), which assess the deleterious effect of that variant. Based on all this information, the variants are extensively filtered to retain only the relevant variants with a potential effect on tumour formation. Filtering excludes intronic regions, except for the AG/GT splice sites, silent changes, recurrent variants that likely are sequencing or homopolymer errors that often occur at VAF <10%, and single nucleotide polymorphisms (SNPs) present in the healthy population. This last step has to be taken with caution since some germline variants are known to predispose to cancer. Finally, it is highly recommended to manually inspect variants in sequence visualization programs such as Integrative Genomic Viewer (IGV; the Broad Institute, Cambridge, MA, USA) to assess the correctness of the annotation, which is especially important for flanking or neighbouring variants, and indels. Note however, that the quality of sequencing and mapping of the reads should always be checked for, to avoid false positives.

This variant annotation forms the basis of the biological classification, i.e., the estimation of the likeliness that a variant contributes to the development or establishment of tumorigenesis. The expert panel decided to use the five biological classes of the ACMG and AMP Standards and Guidelines published by Richards and colleagues [8], even though these guidelines were originally described for germline variants. These entail the classes Pathogenic, Likely Pathogenic, Variant of Unknown Significance (VUS), Likely Benign, and Benign. Irrespective of their clinical consequence, all known driver variants end up in the Pathogenic class (e.g., c.1799T>A; p.(Val600Glu) in *BRAF*) while variants with a significant minor allele frequency (MAF) in the population, also known as polymorphisms or SNP’s, are classified as Benign (e.g., c.215C>G; p.(Pro72Arg) in *TP53*). However, since the classification is not that clear for many other variants, the expert panel aimed to develop a biological classification workflow to streamline and harmonize this currently very subjective process. Variants classified in a biological category will then be further analysed for their clinical impact.

### 2.3. Biological Variant Classification Workflow

The proposed ComPerMed workflow for biological classification of somatic variants is illustrated in Figure 1. We will elaborate below on the reasoning behind each step.

When starting from a Vcf file, the first step of the biological classification workflow is a technical filtering step to select the high quality variants, usually performed by filtering out the unreliable changes (Figure 1-Box 1). Filtering should be done against variants generated by technical artefacts including stutters in repeat sequences, alignment errors and sequencing artefacts. A hallmark of such erroneous variants is that these are typically found in at least 10% of samples, not including variants in hotspot or variants previously reported in the literature [22]. These technical artefacts can be panel- and/or method-specific and thus have to be defined by each laboratory individually. Secondly, only variants that alter the protein sequence in any way (missense, nonsense, in-frame or frameshift indels and splice variants) have to be retained. Therefore, all variants located in introns, except for the AG/GT splice sites, as well as stand-alone synonymous changes are filtered out. Even though some of these changes might result in an alteration of the protein function, insufficient tools or information is currently available to assess this impact. The final filtering steps are related to the VAF and the read depth (coverage) of a variant. The VAF typically is validated down to 5% with at least several hundred reads at that position, indicating that the variant will be detected with high sensitivity and specificity if at least, e.g., 25 variant reads are present in a total of 500 reads. Again, both thresholds are lab-specific and require firm validation. However, these thresholds can be reduced for specific variants that need to be detected at low frequencies (e.g., *BRAF* c.1799T>A; p.(Val600Glu) in hairy cell leukaemia) but an extensive validation is needed here as well. Pathogenic and Likely Pathogenic variants that are filtered out due to a coverage or VAF below the lab-specific thresholds can, however, be considered for re-analysis with an alternative method.

Variants that pass the above-mentioned technical filtering steps are then checked for their presence in the healthy population databases (Figure 1-Box 2). It was agreed by the ComPerMed expert panel that the gnomAD database, an extension of the Exome Aggregation Consortium (ExAC), was the most comprehensive and curated data repository containing data from more than 125,000 exomes and 15,000 genomes. Moreover, gnomAD reports ethnic-based MAFs, which our workflow uses to assign a variant as (Likely) Benign if the MAF is higher than 0.1% in any of the ethnic populations in which at least 2000 alleles were analysed, e.g., the variant c.4796C>A; p.(Pro1599His) in *ALK* has a MAF Total of 0.02% but the MAF in the African population is 0.25% with an interrogated allele number of 24,968, thereby assigning it as a Likely Benign variant. We thus relax the conservative value of 1% that is based on whole population-based data. Variants with an ethnic-based MAF >0.1% and <1% are classified as “Likely Benign” while those ≥1% as “Benign”. The threshold to discriminate between both classes can be defined by each laboratory individually as these variants will not be included in the clinical report. Note that (Likely) Benign variants will typically yield VAFs close to 100% (homozygous) or around 50% (heterozygous) as these are germline variants. Somatic variants typically have VAFs <50% due to the heterozygous state and the presence of contaminating healthy cells in the sample but can also have VAFs close to 50% or even >50% because of genomic amplifications or loss of the wild-type allele. Importantly, as gnomAD might use alternative gene transcript ID’s compared to those used by annotation programs, clear-cut polymorphic variants can be missed. Therefore, gnomAD-negative variants with VAF’s close to 50% and 100% should be checked in dbSNP (https://www.ncbi.nlm.nih.gov/snp/) as well with critical review of the outcome since dbSNP might include (Likely) Pathogenic variants that were not yet curated. 

The remaining pool of somatic variants then needs to be further classified for their potential tumour pathogenicity. We first classified these variants for which sufficient evidence exists that the corresponding amino acid change is a driver in cancer (e.g., EGFR p.(Leu858Arg) or JAK2 p.(Val617Phe)). For that, a Consensus Pathogenic Variant (CPV) list was established for clinically relevant changes in solid and myeloid tumours (Table 1 and Table 2, respectively).

Both CPV lists were edited via a consensus of at least six NGS experts in solid or myeloid malignancies. Initially, both lists are restricted to the driver variants (hotspots) in those proteins for which the ComPerMed Test Level 1 and 2A (Appendix A) was applicable [23]. In a later stage, both lists will be extended with additional genes and variants. Proteins with amino acid variants present in the CPV lists are thus classified as “Pathogenic” (Figure 1-Box 3). Examples are *BRAF* p.(Val600Met) (V600M) and *SF3B1* p.(Lys666Asn) (K666N). Of special note is that the conversion of the three letter amino acid code to the one letter code should be performed correctly in order to check the CPV lists. Several online tools for this conversion are available and should be consulted in case of doubt. Frequent erroneous conversions are A for Asp or Asn, G for Glu, and T for Trp. However, this issue will be solved by the implementation of bioinformatic classification and interpretation software, which is available from several companies now.

Variants that are absent from the CPV list will, by default, not end up in the ”Pathogenic” class and need further classification. We discriminated those variants by their functional impact. Variants that result in a clear loss-of-function (LoF), i.e., frameshift, nonsense or AG/GT splice site variant (Figure 1 Box 4) will most likely result in a weakened or null allele. If the LoF variant is located in a tumour suppressor (Ts) gene, it is likely to result in haploinsufficiency of that Ts gene, irrespective whether it is present in the last exon, which is a hallmark for cancer (with some exceptions like *RB1*). Therefore, these variants should be classified as “Likely Pathogenic“ (e.g., *TET2* c.5609C>G; p.(Ser1870*); *TP53* c.445dup; p.(Ser149Phefs*32)). Since the driver status for most LoF variants in Ts genes have not been demonstrated through functional assays, we choose not to classify these as “Pathogenic“. On the other hand, if a clear LoF variant is present in an oncogene (e.g., *KRAS* c.343G>T; p.(Gly115*)), it likely will not play a key role since oncogenes need to be activated or overexpressed. Consequently, these variants end up in the “Variants of Unknown Significance (VUS)“ class. Alternatively, non-evident LoF changes, i.e., missense variants or in-frame indels, in Ts- or oncogenes could result in a partial loss or activated allele, and therefore, will be re-directed to a Scoring Table (Figure 1-Box 5). This table will guide each non-evident LoF variant towards a classification as “Likely Pathogenic“ or “VUS“, based on data available in curated databases or in peer-reviewed publications. We compiled a list of Ts- and oncogenes based on information in OncoKb (http://oncokb.org/cancerGenes), the TSGene 2.0 list (https://bioinfo.uth.edu/TSGene/), Vogelstein et al. [24], and IntOGen (https://www.intogen.org) (Appendix A). For the few genes that are reported as Ts gene and oncogene, a selection was made for the tumour type in which it was requested to be screened for.

The Scoring Table consists of four parameters for which a score between −1 and +2 can be obtained (Table 3). The total score will define the class of each variant with a total score of ≥2 resulting in a “Likely Pathogenic“ classification and a score <2 in a “VUS“. 

The first parameter of the Scoring Table is the total number of entries of the specific amino acid change in the Catalogue of Somatic Mutations in Cancer (COSMIC; https://cancer.sanger.ac.uk/cosmic/), preferably the most recent version. Today, COSMIC harbours 6 million manually curated somatic coding variants related to human cancers, with a focus on the census of cancer genes [25]. Depending on the number of entries, a score is assigned to each variant. The threshold per score depends on whether the variant is detected in a solid or a haematological tumour. Indeed, the number of solid tumour samples in COSMIC is much higher compared to haematological samples, which produces a bias towards higher numbers in solid cancer samples. These thresholds were set based on testing of many different variants. To obtain a score of “+2” the threshold was set at a total of 50 entries for solid (e.g., *EGFR* c.2303G>T; p.(Ser768Ile): 265 entries), and 10 for haematological tumours (e.g., *SF3B1* c.2219G>A; p.(Gly740Glu): 15 entries), irrespective of the tumour type in which the variant was detected. A score of “0“ was given if the number of entries was below 10 for solid (e.g., *KRAS* c.169G>A; p.(Asp57Asn): seven entries), or below five for haematological cancers (e.g., *TET2* c.2464A>C; p.(Thr822Pro): two entries) (Table 3). Intermediate numbers were given the score of “+1” (e.g., *BRAF* c.1790T>G; p.(Leu597Arg): 35 entries; or *SRSF2* c.320C>A; p.(Pro107His): seven entries). Our reasoning for the high weight of this parameter was that a tumour-related variant is expected in a larger number of samples compared to variants that occur simply as passengers, with minimal effect on tumour development.

The second parameter relates to the in silico prediction tools SIFT and MutationTaster. SIFT (Sorting Intolerant From Tolerant) determines if an amino acid substitution is deleterious to protein function mainly based on protein conservation with homologous sequences and the severity of the amino acid change (http://sift.bii.a-star.edu.sg/www/SIFT_seq_submit2.html). MutationTaster evaluates DNA sequence variants for their disease-causing potential using a battery of in silico tests, including Polyphen, to estimate the impact of the variant on the gene product (http://mutationtaster.org/). The outcome of both prediction tools is often presented in the annotation files. The score of “+0.5“ is obtained only if both tools predict a damaging or deleterious effect, thereby restricting its impact on the total score.

Parameter 3 entails whether the variant has been reported as tumorigenic in functional studies. For that, we advise to use PubMed (https://www.ncbi.nlm.nih.gov/pubmed), Jax-CBK (https://ckb.jax.org/), MD Anderson Personalized Cancer Therapy (https://pct.mdanderson.org), and My Cancer Genome (https://www.mycancergenome.org/) as the databases to check. If a functional relation with cancer has been reported in any of these databases a score of “+0.5“ is attributed to that variant.

Finally, Parameter 4 looks for a detrimental role described for that variant in CIVIC (https://civicdb.org/home), ClinVar (http://www.clinvar.com/), OncoKb (http://oncokb.org/#/), and VarSome (https://varsome.com/). If present as (Likely) Pathogenic in at least one of these databases in a somatic framework, again a score of “+0.5“ is added. Of special note is that the search for each variant is dependent on the interrogated database, performed at the protein level using the three- or one-letter amino acid code (e.g., Ala316Glu or A316E), or at the DNA level.

After carefully checking the above-mentioned databases the sum of the scores of the four parameters provides the total score. In principle, if the number of COSMIC entries of a particular amino acid variant is high (≥50 for solid and ≥10 for myeloid tumours), the scoring table will always yield the class Likely Pathogenic, irrespective of the outcome of the other parameters. On the other hand, if the number of entries for a specific variant in COSMIC is low (≤10 for solid and ≤5 for myeloid) it will always be classified as a VUS since the sum of the other three parameters cannot reach the final score of two. However, careful analysis of Parameters 2 to 4 is necessary for every variant that enters the Scoring Table irrespective of the number of entries in COSMIC. Strong evidence for a (Likely) Pathogenic role might have been reported in one of the databases, which then could overrule a VUS outcome of the Scoring Table to a higher class. It is expected however that only very few variants will initially fall into this group, but the future availability of many more functional studies will yield more “overruling” variants from this proposed Scoring Table, which then might need revision.

For variants in genes not present in the CPV lists we score those also via the Scoring Table with the addition that they can be classified as Pathogenic if the maximal score of +3.5 is obtained. The overruling possibility can be applied here as well for variants of solid-proven pathogenicity, which need to be discussed by the expert panel.

Of special note is that the classification via the Scoring Table can be easily automated through validated in-house developed or commercial software. Several companies are already developing such a tool, which can be adapted towards selective workflows including the one proposed here.

### 2.4. Exceptions to the Workflow

Notably, each workflow comes with exceptions. A small set of genes and specific variants should not be classified via the general workflow and scoring table, since these will result in an incorrect classification. Our list with exceptions will need to be regularly updated by the expert panel. The current exceptions are listed below.
The Ts gene *TP53* is an exceptional gene because of the plethora of variants detected in many tumour types affecting almost every position of the p53 protein. Therefore, we advise to use two dedicated *TP53* databases to assess variant pathogenicity. The International Agency for Research in Cancer (IARC) *TP53* database (http://p53.iarc.fr/) compiles various types of information on human *TP53* variations in relation to cancer [26]. The second database, Seshat (http://vps338341.ovh.net/), can be used for (predicted) functional consequences of protein changes. The tumour-related outcome is presented in the downloadable Summary report. The consensus class indicated by both tools will be used for *TP53* variant classification. However, in comparison with the ERIC recommendations [27] that classify the −2, −1 and +1, +2 exon flanking splice variants as Pathogenic, we mark them as Likely Pathogenic since these variants are actually not different from frameshift variants that are also classified as Likely Pathogenic by ERIC. Secondly, synonymous changes that are predicted to affect splicing are classified as Pathogenic by ERIC. Importantly, synonymous changes in P53 should also be checked for a detrimental functional effect in both *TP53* databases.The *BRCA1* and *BRCA2* Ts genes are specifically analysed for variants in gynaecological tumours of the ovary and endometrium, as well as in cancers from breast, pancreas and prostate. In these cases, clear LoF variants (nonsense, frameshift, splice sites) are always classified as Pathogenic, instead of Likely Pathogenic if classified via the ComPerMed workflow. Notably, LoF variants in the last exon as well as all other somatic variants need to be checked for their pathogenicity in different online databases including ARUP (http://www.arup.utah.edu/database/BRCA/), InterVar (http://wintervar.wglab.org/), ClinVar (https://www.ncbi.nlm.nih.gov/clinvar/), Enigma (https://brcaexchange.org/) and LOVD (https://databases.lovd.nl/shared/genes).Sequencing stutters of short tandem repeats (STR), including homopolymers, often occur as sequencing errors that are present in Vcf files at low allele frequencies, typically lower than 5%. However, a true change at an STR site can be discriminated from a stutter if the VAF is significantly higher than the observed stutter error rate present in most samples. The VAF of each STR variant thus has to be checked and if higher than a validated lab-specific threshold, it should be regarded as a true event. This threshold has to be defined by each lab since it can be method or analysis-specific. Each true STR change has to follow the standard classification workflow. As the prime example, the frameshift c.1934dup in the Ts gene *ASXL1* is often seen as a stutter error in many NGS workflows at VAFs up to 10% but can be also found as a true variant in AML samples with VAF’s above the lab-specific threshold, thus classifying it as Likely Pathogenic.Splice site variants should be restricted to the −2, −1 and +1, +2 exon flanking positions, which harbour the AG/GT consensus splice motif, except for the *MET* exon 14 and *BRCA1* and *BRCA2* splice regions that should be analysed more broadly. All splice site variants will be considered as loss-of-function variants (Likely Pathogenic class) even though it might result in an in-frame exon(s) deletion because loss of at least one exon will most likely functionally harm the protein as well.Splice site variants in exon 14 of MET have to be seen as a CPV and thus are biologically classified as Pathogenic.Out-of-frame indels in exon 9 of *CALR,* including the typical type I and type II mutations, as well as out-of-frame insertions in exon 11 of *NPM1* should not be treated as frameshift variants but as Consensus Pathogenic Variants (CPVs). Therefore, they are classified as Pathogenic.Somatic in-frame indels in the bZIP domain of *CEBPA* should be regarded as Likely Pathogenic.Finally, population-specific very rare benign variants can be distinguished from true somatic variants by their presence in at least three region-specific samples, with VAFs close to 50%, irrespective of the tumour content or tumour type. Consultancy of neighbouring NGS labs is advised and follow-up of such Likely Benign variants is warranted.


All classified variants should be collected in a lab-specific internal database generating a fast and standardized categorization of previously encountered variants. It also allows statistics on their frequency and easy follow-up in case of a required class switch. We anticipate comparing the internal databases of clinical laboratories and to build a common Belgian ComPerMed repository for an extended national standardization.

### 2.5. The Clinical Report

The clinical report should be as concise as possible and needs to contain the information required by ISO 15189 for accreditation of medical laboratories. Lengthy reports of more than five pages are discouraged. The content of the clinical report is described in more detail in the BELAC documents point 4.11.4 Clinical report at page 22 (https://economie.fgov.be/sites/default/files/Files/Publications/files/Belac-NL/2–405NGS-NL.pdf). It should include at minimum the order number, name of the patient, laboratory and physician. The medical information should contain the primary tumour type and histology as well as the clinical information and specific request, if any. Table 4 lists the required information that was agreed upon by the expert panel, including examples for each. In addition to the sample ID, several sample-specific parameters need to be provided as well.

The section ‘Test results’ should include the gene symbol (e.g., *EGFR*), the variant according to the HGVS nomenclature at coding (c)DNA (e.g., c.2573T>G) and protein (e.g., p.(Leu858Arg)) level, as well as the VAF. Though the use of the three letter amino acid code is obliged, the one letter annotation (e.g., L858R) can be added since this shorter form is generally more familiar to clinicians. The genomic position of each variant should not be included. Instead, the NM reference number (NCBI) with version (e.g., *EGFR* NM_005228.5) has to be available in the lab guide or on the report at least for each reported gene. VUS classified variants must be clearly separated from the Pathogenic and Likely Pathogenic variants and preferably are added at the end of the report or as annex. We recommend to add a disclaimer for any VUS so that misinterpretation is excluded. In any case, it is imperative that oncologists/haematologists clearly understand the meaning of these VUS changes. Benign and Likely Benign variants must not be reported. Importantly, the “Test results” section should also indicate which regions/exons/genes could not be interpreted or analysed due to biological or technical reasons, to inform the clinician of any failed amplicons.

In the next section ”Conclusion and interpretation” the clinical interpretation of each Pathogenic and Likely Pathogenic variant together with its clinical tier level needs to be provided. For every biologically classified variant a clinical class needs to be attributed. We propose to use the four-tiered ACMG/AMP guideline system described by Li and colleagues [12], which is regarded as the standard for clinical classification. These four classes are: Tier I (level A and B evidence) for variants of strong clinical significance in the tumour type investigated; Tier II (level C and D evidence) for variants of potential clinical significance; Tier III for variants of unknown clinical significance; and finally, Tier IV for Benign or Likely Benign variants. The clinical classes Tier I, II and III can harbour biologically classified Pathogenic or Likely Pathogenic variants, which all should be included in the clinical report. A VUS is always Tier III while Tier IV only contains (Likely) Benign variants. The clinical Tier III, in case of a VUS, as well as Tier IV class should not be reported in the ‘Conclusion and interpretation’ section. Clinical trials can be included here as well.

The clinical description of Tier I, II and III variants always need to be seen in relation to the tumour type it was detected in, and should be substantiated by firm publications and up-to-date clinical databases. Moreover, the clinical impact of each variant, i.e., its diagnostic and prognostic value, and its sensitivity or resistance to therapy, has to be added in sufficient detail with inclusion of unequivocal referencing (e.g., Brown et al. 2015; J. Mol. Diagn. or PMID: 31138260). In case of absence of a variant in a gene that confers resistance to therapy, it is also important that this absence is clearly mentioned, e.g., no *KRAS* variant in colorectal adenocarcinoma. However, this section should be as concise as possible not to overload the physician with superfluous or irrelevant information.

Clinical classification of variants is often not straightforward and is, to a variable extent, amenable to subjective interpretation. A common point of discussion is how to make the difference between Level B evidence from well-powered studies (Tier I) and Level C evidence from multiple small studies (Tier II). Second, how to interpret prognostic markers in haematological malignancies e.g., Likely Pathogenic variants in *ASXL1*, *TET2*, *DNMT3A*, *RUNX1*. How to define ‘sufficient evidence’ to rank them all as Tier I? Finally, combinations of variants might also affect the clinical outcome. These include resistance variants, which are detected with or without an activating variant. For instance, EGFR c.2239_2240delinsCC; p.(Leu747Pro) confers resistance to TKI treatment in patients with an activated EGFR pathway [28,29] and based on the Scoring Table it is classified as Likely Pathogenic. However, the clinical interpretation is highly dependent on the co-occurrence of an activating *EGFR* variant (Tier I) or not (Tier III). Double variants in *CEBPA* is another prime example in this category [30,31].

The section ‘NGS method’ should describe in sufficient detail the method, including the type of sequencer, the targeted genes and their regions or hotspots of the panel (e.g., *BRAF*: exons 11 and 15; *DNMT3A*: all coding exons), the reference genome used (e.g., Hg19), the validated coverage (e.g., >500×) and VAF (e.g., >5%) thresholds, and all regions or hotspots that consistently fail in the assay (e.g., *DNMT3A*: exon 6 AA165–184; *JAK2*: K539). Moreover, a disclaimer has to be added that the test cannot discriminate between somatic and germline variants. Note that the method section can also be consulted in more detail in the versioned lab guide.

Finally, the report should name the person(s) who interpreted and validated the clinical report as well as the date the report was validated, in agreement with the ISO 15189 norm.

## 3. Discussion

Several studies already proposed guidelines for the clinical interpretation of somatic variants in cancer, which is a key step in the process towards efficient clinical patient management [9,10,11,12,13,14,32], of which some were critically compared [11,33]. However, this clinical interpretation vastly relies on the preceding biological variant classification, i.e., is the biological and/or biochemical evidence sufficiently high to claim that the variant effectively contributes to tumour formation, independent of the tumour type. Even though every diagnostic laboratory performs this critical step, not a single paper reports on a detailed somatic variant classification pipeline and proposes a workflow for improved harmonization. However, several groups describe their classification pipeline for germline variants in inherited diseases, including cancer [34,35,36,37,38,39], which are all based on the ACMG guidelines reported in 2015 [8] and 2017 [12]. For the biological classification of somatic variants the Houston Methodist Variant Viewer (HMVV) provides easy access to the required biological information of each variant but does not per se harmonize its final outcome [40]. Very recently, the Spanish guidelines for the interpretation of NGS variants in myelodysplastic syndrome and chronic myelomonocytic leukaemia has been reported by GESMD [16].

The expert panel of ComPerMed was installed with the aim to streamline the complete process from variant detection up to reporting. We decided to use the five biological classes proposed by the ACMG too, even though these were meant for germline variants [8], i.e., Pathogenic, Likely Pathogenic, VUS, Likely Benign, and Benign. To uniformly classify variants in either one of these biological classes, a consensus ComPerMed workflow was installed (Figure 1). From the Vcf files, we first filtered out the technical errors and common population variants. Then, we checked the variants in the CPV lists, which included the mutational hotspots of the selected ComPerMed genes to be minimally analysed in solid and myeloid tumours. If present, these were scored as Pathogenic. From the remaining variants, the clear LoF variants were classified based on their presence in a Ts gene (Likely Pathogenic) or oncogene (VUS). Even though nonsense-mediated mRNA decay (NMD) is often not induced when the LoF variant is located in the last exon, the ComPerMed panel decided to classify those variants in Ts genes also as Likely Pathogenic because the impact of NMD on cancer genes with premature stop codons is complex and therefore, currently unpredictable [41]. Moreover, a pathogenic effect of such an event has been proven for Ts genes including *ASXL1*, *BRCA1* and *BRCA2*. Finally, a potential dominant-negative effect of the mutant protein can inhibit the function of the wild-type allele product thereby stimulating haploinsufficiency. The non-evident LoF variants were subjected to the four parameter Scoring Table in order to discriminate them between the classes Likely Pathogenic and VUS. All parameters need to be assessed for every variant as new strong evidence-based information can still overrule the outcome of this pipeline. Notably, a list of exceptions to the pipeline was compiled to adjust for an inaccurate outcome of some gene-specific variants.

The guidelines for biological classification we propose here differ from those described by ACMG [12] and GESMD [16] at some critical points:
(1)The minor allele frequency (MAF) threshold to assign a variant as a polymorphism (class Likely Benign or Benign) was set at 1% by ACMG and GESMD, which is especially important in the absence of paired normal tissue. We have lowered this threshold to 0.1% because of the much higher number of data since 2015, and the curation of population databases thereby minimising the contamination of somatic tumour variants. Moreover, this threshold is valid for ethnic-specific MAFs with at least 2000 alleles investigated [42], which can be consulted in gnomAD. Note however, that this MAF threshold can be influenced by the targeted NGS method employed [43]. Finally, variants in *ASXL1*, *DNMT3A* and *TET2* with VAFs below 10% can be associated with Clonal Hematopoiesis of Indeterminate Potential (CHIP) [44] and thus should be interpreted with caution as their presence alone is no evidence for the presence of malignancy;(2)ACMG advises to use the genomic coordinates of variants to be able to query genomic databases and not to depend on transcripts that are prone to changes. We recommend to use the HGVS nomenclature with reference to the transcript ID with the NCBI accession number of the main transcript, with version (e.g., *BRAF* NM_004333.5). We anticipate to change to the Locus Reference Genomic (LRG) record (http://www.lrg-sequence.org/) as it contains a stable reference sequence. So far, not all genes acquired an LRG number, and since most annotation programs and databases do not yet include the LRG transcript numbers, we did not make this switch yet;(3)For splice variants, we only consider the intronic −2, −1 (AG) and +1, +2 (GT) consensus splice positions, except *MET* exon 14 and *BRACA1/2*. ACMG and GESMD also evaluates intronic and exonic variants in the proximity of the splice sites, which are subjected to in-silico splice prediction tools. However, because of the low specificity of these tools and the inherent requirement for functional confirmation, we are not in favour of this option;(4)We classify the LoF variants in the last exon in the same way as those in preceding exons. GESPD requires the further evaluation of these changes.


This ComPerMed workflow is mainly generated with the aim to be as objective as possible. For that, the indicated threshold values of all steps need to be rigorously applied. If a variant has an ethnic-specific MAF of 0.08, it cannot be classified as (Likely) Benign as the MAF is lower than 0.1%. However, if detected in the lab in at least three samples at VAF close to 50% it can still obtain the (Likely) Benign class, which thus can differ between labs. Also, the exact amino acid change and not a similar one has to be searched for in the CPV list and indicated databases. It is therefore crucial to make the correct conversion from the three letter to the one letter amino acid code as different databases require the input of either of both. For genes not yet present in the CPV list, the lab can add the hotspot positions of that gene taking into account strict selection criteria for each variant. The cancer hotspot database https://www.cancerhotspots.org/#/home can be of use as a good starting point [45]. Finally, the lab-specific coverage and VAF thresholds should be strictly applied. However, for some variants these thresholds can be relaxed if sufficiently validated. For example, a type I *CALR* 52 bp deletion variant with coverage below the threshold can be considered as a correct call. Similarly, detection of a BRAF V600E in hairy cell leukaemia often requires a VAF below the general threshold of the panel. Thorough validation of that position can make this detection possible. In general, critical interpretation of each step of the pipeline is always required since it is expected that some variants won’t fit our standard pipeline. Any deviation needs to be sufficiently motivated and discussed by the ComPerMed expert panel.

## 4. Conclusions

We provide for the first time a detailed workflow for the biological classification of somatic variants in solid as well as haematological tumours. This workflow was generated in order to harmonize the reporting of targeted NGS variants in diagnostic laboratories in Belgium under ISO 15189 accreditation. However, it can also be implemented in other countries or serve as a scaffold for implementation. In any case, it will foster discussions on this so far neglected topic, which will be used to improve the workflow in its future versions.

## Figures and Tables

**Figure 1 cancers-11-02030-f001:**
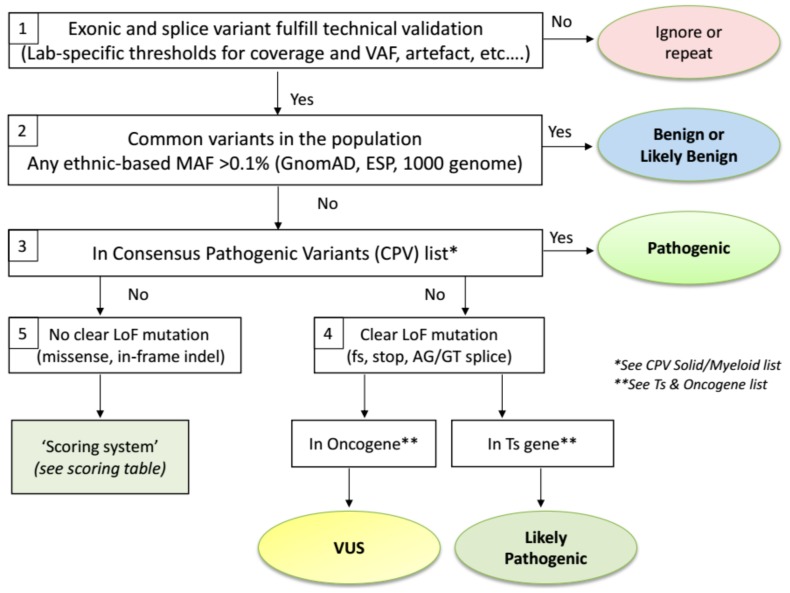
ComPerMed workflow for the biological classification of somatic variants.

**Table 1 cancers-11-02030-t001:** Consensus Pathogenic Variant (CPV) list of the ComPerMed genes selected for screening in solid tumours.

Gene	Transcript ID	Hs1	Hs2	Hs3	Hs4	Hs5	Hs6	Hs7	Hs8	Hs9	Hs10	Hs11	Hs12
ALK	NM_004304.4	F1174L	R1275Q										
BRAF	NM_004333.5	G469A/E/R/V	D594G/M	T599-K601 if-del/ins	V600E/K/M/R	K601E							
BRCA1	NM_007294.3	all clear LoF variants (nonsense, frameshift, splice site)											
BRCA2	NM_000059.3	all clear LoF variants (nonsense, frameshift, splice site)											
EGFR	NM_005228.4	G719A/C/S	ex19if-del/ins	ex20 if-ins	T790M	C797S	L858R	L861Q					
ESR1	NM_000125.3	K303R	E380Q	V392I	S463P	V533M	V534E	P535H	L536H/P/Q/R	Y537C/N/S	D538G		
GNAS	NM_000516.5	R201C/H											
H3F3A	NM_002107.4	K28M	G35R/W										
HRAS	NM_005343.3	G12C/D/S/V	G13C/D/R/S/V	Q61H/K/L/R									
IDH1	NM_005896.3	R132C/G/H/L/S											
IDH2	NM_002168.3	R140L/Q/W	R172K/M/S										
KIT	NM_000222.2	ex8	ex9	ex11	ex11	ex11	ex11	ex11	ex13	ex13	ex14	ex17	ex17
		D419 if-del	S501-F504 if-ins	K550-V560 if-indel	W557G/R	V559A/D	V560D	L576P	K642E	V654A	T670I	D816H/V/Y	N822K
KRAS	NM_004985.4	G12A/C/D/F/R/S/V	G13C/D/R/S/V	A59T	Q61H/K/L/R	K117N	A146T						
MET	NM_001127500.3	ex14 skipping											
NRAS	NM_002524.4	G12A/C/D/R/S/V	G13C/D/R/S/V	A59T	Q61H/K/L/R	K117N	A146T						
PDGFRA	NM_006206.5	S566_E577 if-del	D842V	D842_I843 if-del	V561D								

Hs: Hotspot; if-del: inframe deletion; if-ins: inframe insertion; _: denotes the exact positions of that change; -: denotes a region in which the change has to be located; LoF: Loss of Function.

**Table 2 cancers-11-02030-t002:** Consensus Pathogenic Variant (CPV) list of the ComPerMed genes selected for screening in myeloid tumours.

Gene	Transcript ID	Hs1	Hs2	Hs3	Hs4	Hs5	Hs6
ASXL1	NM_015338.5	none					
CALR	NM_004343.3	ex9of-del	ex9of-ins				
CEBPA	NM_004364.3	none					
CSF3R	NM_156039.3	T618I					
DNMT3A	NM_ 175629.2	R882C/H					
EZH2	NM_004456.4	Y646F/H/N/S					
FLT3	NM_004119.2	ex14if-dup	D835A/E/H/V/Y				
IDH1	NM_005896.3	R132C/G/H/L/S					
IDH2	NM_002168.3	R140L/Q/W	R172K/M/S				
JAK2	NM_004972.3	ex12 if-del/if-dup	V617F				
KIT	NM_000222.2	*see CPV Solid list*					
MPL	NM_005373.2	S505N	W515any ms				
NPM1	NM_002520.6	ex11of-ins					
RUNX1	NM_001754.4	none					
SETBP1	NM_015559.3	D868N	G870S				
SF3B1	NM_012433.3	E622D	R625C/H	H662Q	K666N/R/T	K700E	G742D
SRSF2	NM_003016.4	P95H/L/R	P95_R102del				
TET2	NM_001127208.2	none					
TP53	NM_000546.5	R175H	Y220C	G245S	R248Q/W	R273C/H	R282W
U2AF1	NM_006758.2	S34F/Y	Q157P/R				
WT1	NM_024426.5	none					

Hs: Hotspot; if-del: inframe deletion; if-dup: inframe duplication; of-del: out of frame deletion; of-ins: out of frame insertion; any ms: any missense variant; none: no consensus pathogenic variants present.

**Table 3 cancers-11-02030-t003:** Scoring Table for the biological variant classification of non-loss-of-function (LoF) variants.

Parameter	Score +2	Score +1	Score +0.5	Score 0	Score −1
Total # of entries of that particular AA change at that position in COSMIC	Solid: ≥50	50 > x > 10	*/*	≤10	*/*
	Hemato: ≥10	10 > x > 5	/	≤5	*/*
In silico prediction tools SIFT and MutationTaster	/	/	Both damaging and deleterious	Other	/
Harmful in functional studies (PubMed, JAX-CKB, MDA, MCG)	/	/	Yes	Not reported	No
Described in at least one genomic db (CIVIC, ClinVar, OncoKb, VarSome)	/	/	As (Likely) Pathogenic	Not described/unknown	As (Likely) Benign

Variants with a score ≥2 will be classified as “Likely Pathogenic“. Variants with a score <2 are classified as “VUS“.

**Table 4 cancers-11-02030-t004:** Sample information required on the report.

Parameter	Example(s)
Sample ID (primary lab)	123-45678
Sampling date	16th January 2019
Date of sample received	17th January 2019
Sample tumoral stage	primary, metastasis
Sample anatomic site	colon, liver, blood, lymph node, …
Sample type	resection, (trephine) biopsy, aspirate, …
Sample procedure	FFPE, fresh frozen, fresh tissue, …
Neoplastic cells (%)	30%, na
Sample quality	disclaimer if sample does not fulfill pre-analytical requirements

na: not applicable.

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
