# Peer review of "Standardization of Somatic Variant Classifications in Solid and Haematological Tumours by a Two-Level Approach of Biological and Clinical Classes: An Initiative of the Belgian ComPerMed Expert Panel"

_cancers, 2019, doi:10.3390/cancers11122030_

Round 1

Reviewer 1 Report

Comments for  “Standardization of Somatic Variant Classifications in Solid and Haematological Tumours by a Two-Level Approach of Biological and Clinical Classes: an Initiative of the Belgian ComPerMed Expert Panel”

In Result 2.1 section, the document from ComPerMed panel meeting in January 2019 was published on BELAC website in July 2019. However, I couldn’t find the document provided in the manuscript from the website. Author should update the website link in the document. For Result 2.1 and 2.2 sections, authors should include the gene list that is proposed by ComPerMed panel for different tumor types. I assume the genes listed in Table 1 and Table 2 in the result section are just a partial list of targeted genes suggested by the Panel. Targeted NGS is an ambiguous term in the current manuscript. There are multiple approaches to design the targeted NGS. The target libraries could be designed for specific variants in a list of genes, or specific exonic regions in a list of genes. Can the whole exon sequencing (WES or Exome Sequencing) data be evaluated in the pipeline proposed by the ComPerMed Expert panel? Different sequencing strategies may require some unique approaches to classify the somatic variants. Result 2.2 section on page 4, IGV is suggested to assess the correctness of variants. This step required users to use mapping result data (mostly in BAM format). It would be good to assess the mapping stats with the BAM files in order to evaluate the sequencing and mapping quality. There are quite a few bioinformatics tools available for this purpose such as PICARD developed by Broad Institute. This may bring up an issue regarding to the member representation of the Expert Panel. Does this expert panel include any bioinformatics expert? On page 6, “Of special note is that the conversion of the 3-letter amino acid code to the 1-letter code should be performed correctly in order to check the CPV lists. Several online tools for this conversion are available and should be consulted in case of doubt. Frequent erroneous conversions are A for Asp or Asn, G for Glu, and T for Trp.” This shouldn’t be an issue with correct bioinformatics tools implemented. Regarding to the scoring table, is there any specific reason or methodology to justify the cutoff value for each score? I am not sure if it is appropriate to assign a score based on the entries from COSMIC. It could be correct to assign a high score to a variant with high number of entries. However, it could be an error to assign a low score to a variant with few number of entries in COSMIC if the mutated variant was found in a rare tumor or on a non-popular gene that few research labs have been working on. Result 2.5 section, it will be helpful to propose a professional formal report template as a general guidance for the NGS clinic lab to generate a report. This report may include multiple sections such as: Testers’ information, sample information, NGS platform and analytical methods, variant report (may consist of different sub-sections based on the biological classifications such Likely Pathologic, Pathologic, Benign, Likely Benign or VUS), corresponding references for each of variants identified, potential target therapy for each variants if exist, and clinic tier levels and so on.

Minor errors:

Figure 1 box 2, “Any etnic-based” should be “Any ethnic-based” The format size and font type are changing in some paragraphs (such as end of page 8 and beginning of page 9)

Author Response

Thanks a lot for the extensive review of our manuscript and for your constructive comments, which I will respond to one by one. Modifications are highlighted in yellow in the revised manuscript.

The BELAC website was somehow modified. The correct one was added.  I apologize for that.

The genes present in Table 1 (solid tumours) and Table 2 (haematological tumours) are the ones proposed by ComPerMed as having sufficient clinical value (diagnostic, prognostic or therapeutic), which thus should be analysed by NGS. I added this information in the text now to make it more clear. However, we recognise that many genes present in the larger tumour panels (or even exomes) or not included in this list. To tackle this restriction, we added the following in 2.3 ‘In a later stage, both lists will be extended with additional genes and variants.’ and in the Discussion ‘For genes not yet present in the CPV list, the lab can add the hotspot positions of that gene taking into account strict selection criteria for each variant. The cancer hotspot database https://www.cancerhotspots.org/#/home can be of use as a good starting point [45].’

Correct. Targeted NGS can include ten to hundreds of genes, and hotspots to full genes. Our classification system should be independent from the targets interrogated. We are currently using it with a TMB panel containing 523 tumour-associated genes. However, it is not meant for genes unrelated to tumours so one can use it for WES data but only for the oncogenes and tumour suppressor genes. We did not specify it in our manuscript since we thought it would be inherent to the clinical field, which is cancer diagnostics.

Indeed, the Bam files are loaded in IGV for manual inspection of variants. Poor sequencing or mapping data can easily be seen here since several recurrent variants will occur within a read and those will be present in different reads as well demonstrating the poor quality of that region. We added that the quality of sequencing and mapping of the reads should be checked for. Note that for the data analysis software of Illumina, Picard makes an integral part of it.

Most members of the expert panel have thorough knowledge in basic bioinformatics and use NGS variant analysis in tumour samples minimally on a weekly basis for several years now. For that, we are always assisted by in house bioinformatics experts.

We agree that the issue of using the correct nomenclature is solved by the implementation of bioinformatic tools, such software tools are not yet implemented in most laboratories so that manual classification is still mainly performed. We added in the text ‘However, this issue will be solved by the implementation of bioinformatic classification and interpretation software, which is available from several companies now.’

Very good point. Indeed, we have extensively discussed the score parameters and thresholds to conclude that the ones proposed in this manuscript will fit most variants (but not all). To counteract this point of critique we added the possibility of ‘overruling the classification based on solid evidence’ (end of 2.3 and Discussion) as you correctly point out with your example. We do realize that this overruling step will introduce some kind of subjectivity but this is preferred over an incorrect classifying at all times.

One template for all clinical reports would be the ideal situation but we don’t see that possible because of technical, professional, LIMS-related, software package-related, or hospital-specific reasons. Therefore, we opted to list the information that should be minimally present on each report as well as the order of this information (Tester, patient, sample, variant, clinical (with references), and technical (method) information, concluded with the signature of the interpreter with the date. We added now the referral to the BELAC document in which this requirement is described in more detail (see page 10, bottom). It is not a template but in our opinion the most acceptable guideline towards harmonization.

Minor modifications: Ethnic was corrected in Figure 1. The type font and size has been corrected by the Cancers editorial office.  

Reviewer 2 Report

The manuscript is a report that compiles the details of a workflow system established by the ComPerMed panel of Belgian experts in cancer diagnostics. Their goal was to harmonize the biological classification and clinical interpretation of somatic variants detected by NGS in cancer. As they claim, this effort has resulted in the establishment of a uniform, two-level classification workflow system (biological + clinical) that should enable high consistency in diagnosis, prognosis, treatment and follow-up of cancer patients. For the clinical level of classification, the authors adhere to what has been previously established by ACMG (doi: 10.1016/j.jmoldx.2016.10.002). Thus, their effort has been focused on harmonizing the biological classification.

The report is clear, comprehensive and well structured. It will be useful for any researcher interested in cancer biology, cancer genomics or oncology.

I would like to make a couple of minor comments, devoted to polishing the manuscript:

There is a good resource when dealing with cancer drivers, which is the IntOGen-mutations platform (http://www.intogen.org/mutations/); it identifies cancer drivers according to specific criteria (doi:10.1038/nmeth.2642). Maybe it could be included along pipeline in order to refine the classification of some variants. The type font and size is different at some points of the manuscript: Page 5, from “… sample but can also have VAFs close to 50%...” Page 8, last paragraph and beginning of page 9 Page 13: https://www.cancerhotspot... Page 14, at Supplementary Materials, “Figure S1…”

Author Response

Thank you for your positive comments and for confirming the high value of this work in the broad field of cancer. Modifications are highlighted in yellow in the revised manuscript.

We aded the IntOGen-mutations link, which was unknown to us. Thanks. The type font and size has been corrected by the Cancers editorial office.

Reviewer 3 Report

The collection of such  data is not easy. For this reason, the paper deserves such a credit for carrying out such a study. Most of analyses are carried out in a very reasonable way. I agree with the finding of the results may be important to the gene varitation detection.

Author Response

Thank you for your positive comment and for correctly judging the difficulty of this task.